# Hibernator-Derived Cells Show Superior Protection and Survival in Hypothermia Compared to Non-Hibernator Cells

**DOI:** 10.3390/ijms21051864

**Published:** 2020-03-09

**Authors:** Koen D.W. Hendriks, Christian P. Joschko, Femke Hoogstra-Berends, Janette Heegsma, Klaas-Nico Faber, Robert H. Henning

**Affiliations:** 1Department of Clinical Pharmacy and Pharmacology, University Medical Centre Groningen, University of Groningen, 9713 GZ Groningen, The Netherlands; c.p.joschko@student.rug.nl (C.P.J.); f.hoogstra-berends@umcg.nl (F.H.-B.); r.h.henning@umcg.nl (R.H.H.); 2Department of Surgery, University Medical Centre Groningen, University of Groningen, 9713 GZ Groningen, The Netherlands; 3Departments of Gastroenterology and Hepatology, University Medical Centre Groningen, University of Groningen, 9713 GZ Groningen, The Netherlands

**Keywords:** hibernation, mitochondria, ischemia-reperfusion, hypothermia, reactive oxygen species, ferroptosis

## Abstract

Mitochondrial failure is recognized to play an important role in a variety of diseases. We previously showed hibernating species to have cell-autonomous protective mechanisms to resist cellular stress and sustain mitochondrial function. Here, we set out to detail these mitochondrial features of hibernators. We compared two hibernator-derived cell lines (HaK and DDT1MF2) with two non-hibernating cell lines (HEK293 and NRK) during hypothermia (4 °C) and rewarming (37 °C). Although all cell lines showed a strong decrease in oxygen consumption upon cooling, hibernator cells maintained functional mitochondria during hypothermia, without mitochondrial permeability transition pore (mPTP) opening, mitochondrial membrane potential decline or decreased adenosine triphosphate (ATP) levels, which were all observed in both non-hibernator cell lines. In addition, hibernator cells survived hypothermia in the absence of extracellular energy sources, suggesting their use of an endogenous substrate to maintain ATP levels. Moreover, hibernator-derived cells did not accumulate reactive oxygen species (ROS) damage and showed normal cell viability even after 48 h of cold-exposure. In contrast, non-hibernator cells accumulated ROS and showed extensive cell death through ferroptosis. Understanding the mechanisms that hibernators use to sustain mitochondrial activity and counteract damage in hypothermic circumstances may help to define novel preservation techniques with relevance to a variety of fields, such as organ transplantation and cardiac arrest.

## 1. Introduction

Hibernating species are well known for their ability to initiate safe metabolic suppression and to resist ischemia and hypothermia, even outside the hibernation season [1,2,3,4,5]. Indeed, during and after natural hibernation, when core temperature drops dramatically to ~4 °C, no organ damage was found [6,7]. Even more interesting is that outside the hibernation season, hibernators withstand iatrogenic damage, such as ischemia/reperfusion injury (IRI) and energy deprivation [8,9,10,11], whereas IRI in humans leads to organ failure as found, for example, in organ transplantation [12] and myocardial infarction [13]. Because of these features, hibernation is a highly interesting model to define new preservation techniques in conditions such as organ transplantation [14,15] and cardiac arrest [16]. Previously, we and others found that cultured cells from a hibernator showed superior survival compared to similar non-hibernator derived cells after hypothermia–rewarming [17,18], suggesting the existence of a cell-autonomous protection mechanism in hibernator cells. In keeping with the many functions that mitochondria fulfil in balancing energy production, providing cell homeostasis and initiating apoptosis, these initial studies showed important adaptations of hibernator mitochondria. 

Tissue damage induced by IRI is generally accompanied with adenosine triphosphate (ATP) deficiencies, excessive formation of reactive oxygen species (ROS), opening of the mitochondrial permeability transition pore (mPTP) and loss of mitochondrial protein quality control [19,20,21,22], collectively denoting dysfunction of mitochondria. With similar observations being made in hypothermia, it can be viewed as a special form of IRI. At lower temperature, mitochondria have been shown to strongly decrease their activity but still produce ROS [23]. This discrepancy in oxygen consumption and ROS formation may result from a mismatch in the activity of the five mitochondrial respiratory complexes; while complex I and II are accepting electrons, complex IV is not able to release them properly, resulting in a loss of mitochondrial ATP production and the production of free radicals. As a result, ATP levels can insufficiently be maintained for normal cellular activity, and mitochondrial ROS (mtROS) damages proteins and/or DNA, both eventually leading to a loss of function or even cell death. Nowadays, a variety of different cell death pathways are known, among which are the well-known controlled apoptosis and uncontrolled necrosis [24]. Clarification of the exact cell death pathway induced by hypothermia is important to find new targets. We suggest that the more recently discovered ferroptosis pathway, a form of regulated cell death initiated after an overload of lipid peroxidation [25], plays an important role in hypothermia. 

Here, we explored the mitochondrial function in hibernator- and non-hibernator derived cell lines during hypothermia and rewarming. Additionally, we sought to determine the cause of cell death by exploring ferroptosis in both hibernator and non-hibernator cells.

## 2. Results

### 2.1. Hibernator-Derived Cells Showed Superior Survival Following Hypothermia Compared to Non-Hibernators

To establish the survival of hibernator-derived cells in hypothermia, we induced hypothermia at 4 °C in four different cell lines, two of which derived from non-hibernators (HEK293, human, and NRK, rat) and two from a hibernator (HaK and DDT1-MF2, both Syrian hamster). Cells were cooled for different time periods and subsequently rewarmed for 2 h at 37 °C (Figure 1). A relatively short hypothermic period of 6 h did not affect cell viability any of those cell lines. However, 24 h of hypothermia resulted in complete cell death in both non-hibernator derived cell lines (Figure 1a,c), whereas the hibernator cells showed (nearly) complete cell survival (Figure 1b,d). Even after an extended cooling period of 48 h, hibernator-derived cells still showed remarkable survival rates.

### 2.2. Hibernator-Derived Cells Maintain Mitochondrial Activity during Hypothermia Compared to Non-Hibernator Cells

Next, we examined mitochondrial activity of cells at normal temperature and hypothermia by measuring state 3 and uncoupled oxygen consumption, mitochondrial membrane potential and mitochondrial ROS production, at normal and hypothermic temperatures (Figure 2a–d).

Interestingly, baseline state 3 respiration levels of the hibernator-derived cell lines at 37 °C were markedly higher compared to non-hibernator cells. At 4 °C, all cell lines showed a comparable relative decline in oxygen consumption, thus resulting in the absolute respiration being higher in hibernator cells compared to non-hibernator cells (Figure 2a,b). To investigate whether the maximum capacity of the respiratory chain differs between non-hibernators and hibernators, we next determined maximal oxygen consumption by uncoupling the mitochondrial membrane using Carbonyl cyanide-p-trifluoromethoxyphenylhydrazone (FCCP) (Figure 2c). Uncoupling showed a similar pattern to state 3 and increased oxygen consumption in the hibernator cells compared to the non-hibernators with a strong decrease upon hypothermia.

As the mitochondrial membrane potential (MMP) is built by complex I to III and drives the ATP production, we analyzed the MMP as a surrogate measurement of mitochondrial activity. Expectedly, hypothermia induced a decrease in the MMP in non-hibernator cells, though it induced a strong increase in hibernator-derived cells (Figure 2d).

To examine whether these mitochondrial differences explain dissimilarities in cell survival during hypothermia, we examined mitochondrial permeability transition pore (mPTP) opening and caspase 3 and 7 activity at 6 h of hypothermia (Figure 2e–f). Whereas hypothermia resulted in an increased mPTP opening in non-hibernator derived cells, mPTP opening was unaffected in hibernator cells. However, mPTP opening in non-hibernator cells did not result in increased caspase activity. More specifically, we found a decrease in caspase activity upon cooling, which was comparable in all four cell lines, suggesting that the observed cell death is not mediated by apoptosis (Figure 2f).

Taken together, our data show hypothermia to induce cell death in non-hibernator cells along with mitochondrial failure, whereas hibernator cells sustain mitochondrial activity during hypothermia without cell death.

### 2.3. Hibernators withstand ROS Damage and Ferroptosis in the Cold

Next, we examined mitochondrial ROS production in the different cell lines at normothermia and hypothermia. Interestingly, while non-hibernator cells showed a substantially lower mitochondrial oxygen consumption at 37 °C compared to hibernator cells (Figure 2c), mitochondrial superoxide production was markedly higher in non-hibernating derived cells compared to hibernator cells (Figure 3a). Further, during hypothermia, MitoSOX fluorescence of all cell lines declined to comparable levels. Contrasting to these decreases in MitoSOX values, lipid peroxidation increased markedly after exposure to 4 °C in non-hibernator cells but remained stable in the hibernators (Figure 3b). Interestingly, the increased lipid peroxidation in non-hibernators, resulting from long-term superoxides exposure, cannot be explained by overproduction of superoxides, as hypothermia induced a strong decrease in the MitoSOX values, with similar levels in hibernator and non-hibernator cells. More likely, the discrepancy between MitoSOX values and lipid peroxidation in hypothermia exposed cells is based on the handling of superoxides by the cells, such as by scavenging. Therefore, we examined scavenging capacity of cells during normothermia and hypothermia by examining lipid peroxidation following exogenous administration of H_2_O_2_. In keeping with sustained handling of ROS in hibernator cells, H_2_O_2_ treatment did not affect malondialdehyde (MDA) levels in HaK cells both in warm and cold conditions (Figure 3c). In contrast, MDA levels in HEK293 cells were unaffected by H_2_O_2_ treatment under normothermic conditions but increased under hypothermic conditions. Next, to explore whether non-hibernator cells lack general anti-oxidant capacity during hypothermia, we added the strong anti-oxidant Trolox during the hypothermic period. However, Trolox treatment did not protect the non-hibernating cells during hypothermia, indicating that cell death in response to ROS formation is not caused by lack of antioxidant capacity. As ferroptosis is based on the accumulation of lipid peroxidation, we blocked the ferroptosis pathway in cooled cells. Interestingly, inhibition of ferroptosis resulted in a strongly increased survival of hypothermic non-hibernator cells, comparable to levels of hibernator-derived cells (Figure 3d).

As glutathione (GSH) is one of the best known cellular anti-oxidants, we examined GSH levels throughout hypothermia. Interestingly, HaK cells increased their GSH levels directly after hypothermia, whereas HEk293 cells did not (Figure 3e). Although both HaK and HEK293 showed a decrease in GSH levels upon cooling, HEK293 cells showed a strong decrease eventually resulting in a total depletion of GSH within 12 h, whereas HaK cells decreased to a stable GSH level for at least 24 h (Figure 3f).

### 2.4. Hibernator-Derived Cells Maintain Mitochondrial Dependent Energy Production in Hypothermia, by Using Endogenous Energy Sources

Besides ferroptosis and oxidative damage, our data show that the hypothermia associated mitochondrial failure leads to energy depletion in non-hibernator cells, contributing to their cold-induced cell death. Conversely, stress survival in hibernator-derived cells appears to be driven by maintaining mitochondria function and integrity. HEK293 cells showed a rapid ATP depletion during and after hypothermia, whereas ATP levels were stable in HaK cells during cooling (Figure 4a). To rule out a potential storage–release mechanism of ATP in hibernators, HaK cells were pre-treated with inhibitors of the mitochondrial oxidative phosphorylation (oligomycin), glycolysis (2DG) or the combination at 37 and 4 °C. Treatment with the combination of oligomycin and 2DG resulted in a rapid decrease in ATP levels followed by cell death, both in cooled and normothermic cells, excluding a storage–release mechanism of ATP in HaK to survive cooling (Figure 4b). However, the addition of either oligomycin or 2DG did not lead to cell death in HaK cells in either normothermic or hypothermic conditions (Figure 4c), demonstrating that HaK cells survive by aerobic or anaerobic energy production, both compensating for each other, under normothermic and hypothermic conditions.

To identify important energy sources in hypothermia, cells were placed in different cell culture media, and survival was assessed. HEK293 and HaK cells were not able to survive in normothermic culture medium depleted from any energy source (Figure 4f). Interestingly, while complete substrate deprivation expectedly resulted in a complete cell death in hypothermic HEK293 cells, full cell survival was observed in hypothermic HaK cells (Figure 4d). These data thus suggest that cooling mobilizes the use of an endogenous energy source in the hibernator originated HaK cells. To establish whether this substrate uses OXPHOS or glycolysis, survival of HaK was assessed after inhibition with oligomycin, 2DG or both. Again, in HaK, inhibition of either glycolysis or OXPHOS did not affect survival, with the combined treatment HAK cells deprived of exogenous energy sources inducing marked cell death (Figure 4e). Unsurprisingly, HEK293 cells were not able to survive in all conditions lacking external nutrients.

### 2.5. Hibernator-Derived Cells do not Thrive on Autophagy

Since an endogenous energy source was implied in survival of hypothermic HaK cells in absence of extracellular energy sources, we examined autophagy as a potential endogenous energy source in both HaK and HEK293 cells in normothermic and hypothermic conditions. We examined LC3B protein expression, a central protein in autophagy existing in the cytosolic inactive (LC3B-I) and conjugated active (LC3B-II) form. An increase in LC3B-II/LC3B-I ratio may thus indicate the autophagy activity. Whereas HEK293 cells showed comparable LC3B-II/I ratios in normothermia and after hypothermia, hypothermia induced a significant decrease ratio in HaK (Figure 5a). Next, we inhibited the autophagy pathway by bafilomycin. Bafilomycin did not affect cell death after hypothermia in HEK293 or HaK cells (Figure 5b,c). Together, these data indicate that hibernator-derived cells do not thrive on autophagy during hypothermia.

## 3. Discussion

Here, we show that cells derived from a hibernating species display superior stress resistance to cooling. Whereas non-hibernator derived cells did not survive cooling (4 °C), cell death was virtually absent in hibernator cells, even after 48 h of hypothermia. We hypothesized this resistance to cell death in hibernator cells is rooted in a differential mitochondrial response. Indeed, we found several mitochondrial parameters to be preserved during hypothermia in hibernator-derived versus non-hibernator derived cells, including increased oxygen consumption and ATP production, absence of mPTP opening and preserved radical scavenging. Overall, these data signify that hibernators have powerful cell-autonomous mechanisms to resist cellular stress.

Although a mild hypothermia can be protective in stress conditions such as hypoxia [26], deep hypothermia is a strong stress model [17,23,27]. Interestingly, all cell types survived a relatively shorter cooling time of 6 h. However, after 24 h both non-hibernating derived cell lines showed complete cell death, which was preceded by a strong decrease in mitochondrial respiration and a collapse of mitochondrial membrane potential (MMP). Notably, hibernator-derived cells showed an increased MMP, indicating hibernator mitochondria do not suffer from inadequacies during hypothermia. Consequently, they preserve ATP levels and limit ROS production, altogether stabilizing cellular functions and cell survival during hypothermia. Conversely, together with a decreased MMP, hypothermic non-hibernator cells showed ATP depletion and mPTP opening, indicating severe mitochondrial dysfunction. In line with our results, mitochondrial dysfunction has been shown before in hypothermic rat brain [28].

### 3.1. Cell Death in Non-Hibernator Cells

Our data imply that cooling-induced cell death in non-hibernator cells is principally caused by ferroptosis, as it was completely abolished by pre-treatment with its inhibitor, ferrostatin. Moreover, despite the mitochondrial failure of cooled non-hibernator cells, no caspase activation was found, implying that cell death is not caused by apoptosis. In keeping with these results, the mitochondrial dependent superoxide production showed a decrease in non-hibernator cells comparable to that of hibernator cells after induction of hypothermia. Nevertheless, cooled non-hibernator cells showed increased lipid peroxidation, which was present even in cells treated with a potent anti-oxidant (Trolox). While we [23] and others [27,29,30] have shown previously that hypothermia induces ROS damage in non-hibernator cells, our data suggest that this is not at the heart of mechanisms causing cell death. Rather, lipid peroxidation, in turn inducing ferroptosis [25], seems to qualify in line with previous studies showing hypothermia to induce ferroptosis [31]. Another mechanism present in cooled cells that may explain ferroptosis is the strong reduction in mitochondrial function. In turn, the observed mitochondrial failure leads to reduced ATP levels during hypothermia, as previously reported in a variety of non-hibernator species [17,32]. Because of this energy depletion, a plethora of energy demanding cellular functions are inhibited. These include not only Na/K pumps, effectuating the cellular membrane potential, but also the synthesis of important enzymes and anti-oxidants, such as glutathione (GSH) [33]. As depletion of GSH leads to increased lipid oxidation, it plays a critical role in activating ferroptosis [34,35]. In line with the added importance of mitochondrial function in hypothermia besides ferroptosis alone, the mitochondrial protector Sul-109 has shown to stabilize ATP levels during hypothermia, leading to increased cell survival [36]. In contrast to this, neutralizing excessive ROS formation with anti-oxidant did not affect cell survival. Collectively, our data suggest that initiation of ferroptosis by ATP depletion following mitochondrial failure constitutes the main mechanism of hypothermia-induced cell death in non-hibernator cells.

### 3.2. Cell Survival in Hibernator Cells

In contrast to non-hibernator cells, hibernator cells showed stable ATP levels during hypothermia [17]. Although hypothermia induced a comparable drop in oxygen consumption as seen in non-hibernator cells, MMP increased after cooling in hibernator cells. Although MMP and mitochondrial ATP production are not necessarily correlated [37], absence of a drop in MMP suggests a better mitochondrial energy production. Interestingly, inhibition of the energy production by either inhibiting glycolysis or OXPHOS revealed that HaK cells were able to compensate for both of these systems completely both in normothermia and hypothermia, suggesting that both pathways are active during cooling. Moreover, when needed, HaK cells seems to recruit endogenous substrate(s) during cooling. Whereas depletion of extracellular substrates induces massive cell death in HaK at normal temperatures, it was without effect on cell survival in cooled HaK cells. As we did not find an increase in autophagy, the nature of this intra-cellular energy source is still elusive. In addition, our data suggest that hibernator cells have superior oxidant defense, as exposure to an exogenous oxidant did not affect cooled HaK cells, in contrast to non-hibernator cells. Together, these data suggest that hibernator cells survive cooling by maintaining mitochondrial functionality. This results in adequate ATP levels, maintaining GSH levels and prevention of excess ROS damage, thus precluding ferroptosis.

In conclusion, failure of mitochondria in non-hibernator cells leads to ATP depletion and accumulation of ROS, ultimately resulting lipid oxidation and ferroptosis. Hibernator cells avoid this primarily by maintaining functionality of mitochondria. A full understanding of the mechanisms hibernator cells use to sustain mitochondrial function and avoid ferroptosis during hypothermia may reveal the specific cellular protection mechanism, which may ultimately lead to the development of novel approaches to limit organ damage in, for example, transplantation, major surgery or cardiac arrest. Beyond these applications, this may possibly even apply to ferroptosis associated stress conditions comparable to hypothermia, such as stroke [38] and acute kidney injury [39].

## 4. Materials and Methods

### 4.1. Cell Culture

Two hibernator-derived (HaK, hamster kidney, Syrian hamster, ATCC CCL-15 and DDT1MF2, ductus deferens tumor, Syrian hamster, A TCC CRL 12051) and two non-hibernator derived (HEK293, human embryonic kidney, ATCC CRL-1573 and NRK52E, normal rat kidney, *Rattus norvegicus* ATCC CRL-1571) were used. Cells were cultured in Dulbecco’s modified Eagle’s medium (Gibco, Gaithersburg, MD, USA) supplemented with 10% fetal bovine serum (Gibco, USA) and 1% penicillin/streptomycin. For experiments, cells were grown on 0.001% poly-l-lysine coated surfaces at 90% confluency.

### 4.2. Used Chemicals

Carbonyl cyanide-p-trifluoromethoxyphenylhydrazone (FCCP, Sigma-Aldrich, St. Louis, MO, USA), oligomycin A (Sigma-Aldrich, USA), 2-Deoxy-D-glucose (2DG, Sigma-Aldrich, USA), Trolox (Sigma-Aldrich, USA), ferrostatin (Sigma-Aldrich, USA) and bafilomycin (Sigma-Aldrich, USA) were used in the study.

### 4.3. Hypothermia, Rewarming and Cell Viability

To perform hypothermia and rewarming experiments, cells were seeded on culture plates and after overnight adherence placed at 4 °C in a standard laboratory refrigerator for different time periods, with or without rewarming by reinstitution of standard cell culture conditions (37 °C). Cell viability was assessed by a neutral red (NR) or trypan blue (TB) assay to quantify the number of living cells. Neutral red assay: Following replacement of normal media by NR media (culture media with 10% FBS and 50 mg/mL NR dye, Sigma Aldrich), cells were lysed, and absorbance was measured at 450 nm using a Synergy 2 Multi-Mode plate reader (BioTek, Winooski, VT, USA).

Trypan blue assay: To assess cell viability using the TB assay, cells were washed with PBS, trypsinized and then centrifuged. The supernatant was discarded and the pellet resuspended in PBS. TB was added to the cell suspension (final concentration of 0.013%), incubated for 10 min at 37 °C and manually counted in a Bürker counting chamber.

### 4.4. Mitochondrial Respiration

Cells were freshly trypsinized and resuspended in Hank’s balanced salt solution (Gibco, USA) supplemented with 25 mM HEPES. Mitochondrial oxygen consumption was measured using the Oroboros equipment. After a stable temperature was reached (37 or 4 °C), cells were suspended in 800 µL of Mitochondrial Respirometry Solution (MiR05: 0.5 mM EGTA, 3 mM MgCl2•6H2O, 20 mM taurine, 10 mM KH2PO4, 20 mM HEPES, 1 g/L BSA, 60 mM potassium-lactobionate, 110 mM sucrose, pH 7.1 at 30 °C). Cells were permeabilized using digitonin (20 µg/mL) to allow entrance of substrates. State 3 respiration was measured in response to the substrates glutamate and malate with an ADP generating system consisting of ATP (500 mM), hexokinase (500 U/mL) and glucose (1 M). Uncoupled respiration was measured in the presence of FCCP (1 µM). Oxygen consumption rates were normalized to the number of cells and expressed as pmol O2/min/1 × 10^6^ cells.

### 4.5. Mitochondrial Membrane Potential

Cells were plated in 96-well dark plates and incubated at 37 or 4 °C. Mitochondrial membrane potential was measured using JC1 (1 μg/mL, Sigma-Aldrich, USA) by quantifying the fluorescence emission shift from green (529 nm) monomers to red (590 nm) aggregates. FCCP (1 μM) was used as uncoupled control. Data are expressed as fold increase in the red/green ratio.

### 4.6. Mitochondrial Superoxides

Cells were cultured in sterile 96-well dark plates until confluent. One hour prior to the experiment, cells were washed with pre-warmed HBSS, and medium was replaced with HBSS. At the same time, for the positive control, antimycin A was added to the positive controls. Afterwards, MitoSOX reagent was added (5 μM, Thermo Fisher, Waltham, MA, USA), and plates were incubated for 90 min at 37 or 4 °C. Fluorescence was measured (excitation/emission: 510/580 nm) using a Synergy 2 Multi-Mode plate reader (BioTek). Antimycin A pre-treated cells were used as positive control.

### 4.7. mPTP Opening

Cells were cultured in sterile 96-well dark plates until confluent and incubated at 37 or 4 °C for 6 h. mPTP opening was assessed using the MitoProbe™ Transition Pore Assay Kit (Thermo Fisher). Cells were loaded with the probe, probe with cobalt or probe, cobalt and ionomycin according the manufacturer’s protocol. After 10 min of incubation (37 °C), fluorescence was measured (ex/em: 495/515 nm) using a Synergy 2 Multi-Mode plate reader (BioTek). Ionomycin was used as positive control.

### 4.8. Caspase 3/7 Activity

Cells were cultured in sterile 96-well dark plates until confluent and incubated at 37 or 4 °C for 6 h. Caspase activity was assessed using the Caspase-Glo^®^ 3/7 Assay (Promega, Madison, WI, USA) according to manufacturer’s protocol, and luminescence was measured using a Synergy 2 Multi-Mode plate reader (BioTek).

### 4.9. ATP

Cells were grown in 12-well plates and incubated for 12 h at 37 or 4 °C. Following addition of Tris-EDTA buffer, cell scraping on ice and boiling for 6 min, ATP was measured with a luciferase assay (Promega) with luminescence measured at 590 nm. Data are expressed as relative fluorescence levels corrected for protein levels, where 37 °C was set to 1.

### 4.10. Western Blot

Cell lysates were obtained using RIPA lysis buffer (50 mM Tris-Cl pH = 8.0, 150 mM NaCl, 1% Igepal Ca 630, 0.5% Sodium Deoxycholate, 1.0% SDS, 0.4% protein inhibitor cocktail, 1 mM sodium orthovanadate, 10 mM NaF, 10 mM B-mercaptoethanol). Protein concentration was measured with a Bio-Rad protein assay on a Synergy H4 plate reader. Samples were loaded to 4%–20% SDS pre-casted gels (Bio-Rad TGX gels) and transferred to a nitrocellulose membrane (Bio-Rad, Hercules, CA, USA). A stain-free picture was taken for later total protein normalization. The membranes were blocked with 5% skimmed milk and incubated with primary (O/N at 4 °C) and secondary (1 h at room temperature) and visualized using SuperSignal (Perkin Elmer, Waltham, MA, USA) on a ChemiDoc MP imaging system (Bio-Rad), normalized for total protein and quantified using ImageLab 6.0 (Bio-Rad). Antibodies used were anti-LC3B (1:1.000 Cell Signaling Technology, Danvers, MA, USA) and goat anti-rabbit peroxidase (GARPO, 1:000 Dako).

### 4.11. Lipid Peroxidation

Cells were incubated for 6 h at 37 or 4 °C or treated with 500 µM H2O2 for 4 h at 37 or 4 °C. After harvesting, lipid peroxidation was quantified by measurement of malondialdehyde (MDA) using the OxiSelect TBARS assay kit (Cell Biolabs, USA) and expressed relative to 37 °C.

### 4.12. GSH Measurement

GSH content was determined using a spectrophotometry-based assay using glutathione reductase and 2-vinylpyridine as described previously [40,41]. Cells were grown in 6-well plates and exposed to our hypothermia/rewarming protocol according to the experimental procedure. Then, cells were washed twice with ice-cold PBS and harvested using a general lyses buffer (25 mM HEPES, 150 mM KAc, 2 mM EDTA, 10 mM NaF, 2 mM PMSF, 1 µg/µL pepstatin, 1 µg/µL leupeptin, 1 µg/µL a-protein, pH 7.5), exposed to a freeze/thaw cycle (3 times), centrifuged (10 min, 12,000 rpm) and supernatant was used. Values were corrected for protein concentration, determined by the BioRad DC protein assay.

## Figures and Tables

**Figure 1 ijms-21-01864-f001:**
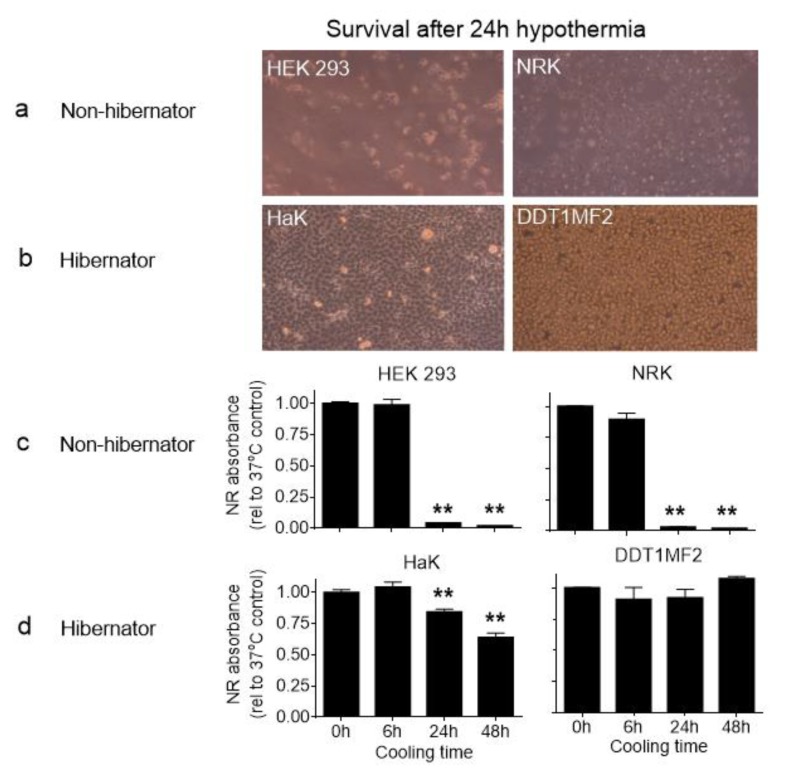
Cell survival after hypothermia in non-hibernator and hibernator-derived cell lines. (**a**) microscopic images (100×) after 24 h hypothermia of the non-hibernation derived HEK293 (human embryonic kidney 293) and NRK (normal rat kidney cells) and (**b**) hibernator derived HaK (hamster kidney) and DDT1MF2 (ductus deferens tumor). (**c**) Cell survival measured as neutral red (NR) absorbance, expressed relatively to non-cooled cells in HEK293 and NRK and (**d**) HaK and DDT1MF2. Data presented as mean ± SD. ** = *p* < 0.01; ANOVA post hoc Bonferroni.

**Figure 2 ijms-21-01864-f002:**
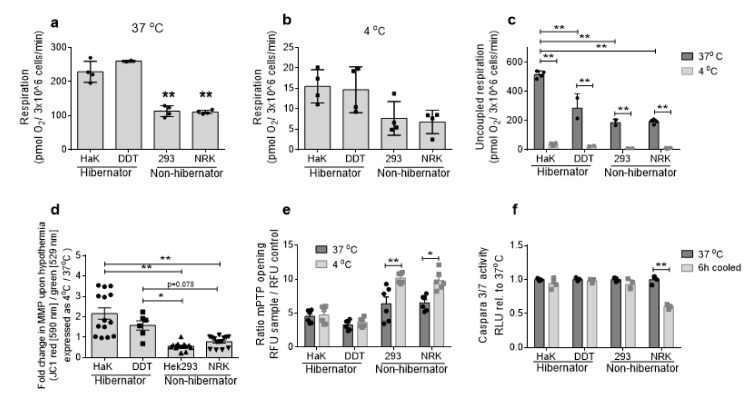
Mitochondrial function during normal temperatures and hypothermia. (**a**) State 3 respiration in digitonin treated cells, energized with malate, glutamate and pyruvate at 37 and (**b**) 4 °C. (**c**) Respiration in Carbonyl cyanide-p-trifluoromethoxyphenylhydrazone (FCCP) treated uncoupled cells at 37 and 4 °C. (**d**) Fold change in mitochondrial membrane potential upon 2 h cold incubation. Shown as fold change in hypothermic versus normothermic for JC1 ratio RFU 590/530 nm. (**e**) Mitochondrial permeability transition pore (mPTP) opening in warm and 6 h 4 °C treated cells. Presented as random fluorescence units (RFU) probe in absence of cobalt divided by cobalt treated controls. (**f**) Caspase 3/7 activity, presented as fold change in 6 h 4 °C treated versus normothermic, random light units (RLU). All data presented as mean ± SD. * = *p* < 0.05, ** = *p* < 0.01; ANOVA post hoc Bonferroni.

**Figure 3 ijms-21-01864-f003:**
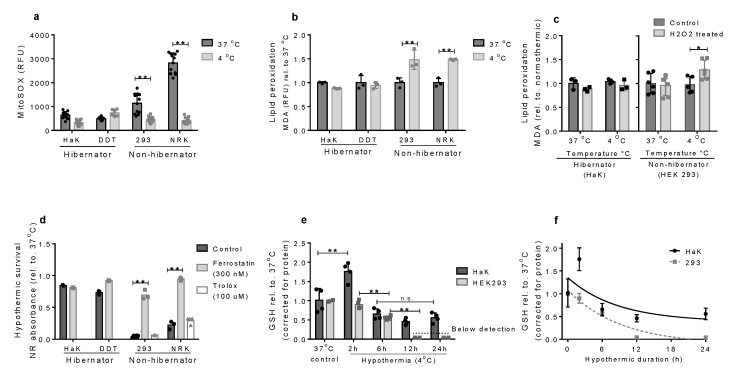
Oxidative damage after hypothermia. (**a**) Superoxide production in normothermic and hypothermic cells, measured by MitoSOX. (**b**) Lipid peroxidation levels induced by hypothermia (4 °C 6 h), measured by malondialdehyde (MDA). (**c**) MDA levels in HaK and HEK293 cells after exogenous H_2_O_2_ exposure in normothermic and hypothermic circumstances. (**d**) Survival after hypothermia expressed as relative NR absorbance to 37 °C control. (**e**) Reduced glutathione (GSH) levels during hypothermia. Expressed as relative levels to 37 °C. (**f**) GSH decline during hypothermia, non-linear fit. The T-test for the area under the curve is statistically different at *p* < 0.001. All data presented as mean ± SD. * = *p* < 0.05, ** = *p* < 0.01; ANOVA post hoc Bonferroni. n.s. = non-significant

**Figure 4 ijms-21-01864-f004:**
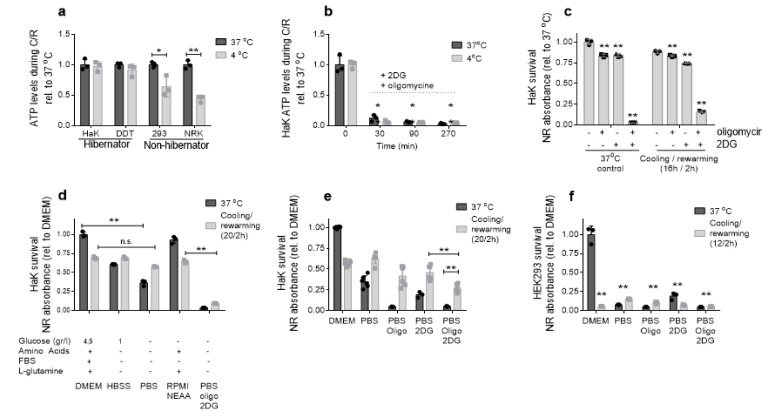
Adenosine triphosphate (ATP) levels and cell survival. (**a**) ATP levels after warm and 12 h cold treatment, expressed as relative to 37 °C. (**b**) ATP levels in HaK cells after 2.5 µM oligomycin and 50 mM 2-deoxyglucose (2DG) treatment over time. (**c**) HaK survival, relative to non-treated 37 °C control. Cells were treated with 2.5 µM and/or oligomycin or 50 mM 2DG and exposed to 16 h hypothermia (hyp) followed by 2 h rewarming. (**d**) HaK survival after cooling and rewarming in different cell culture media: Dulbecco’s minimum essential medium (DMEM), Hanks’ balanced salt solution (HBSS), Roswell Park Memorial Institute (RPMI) and phosphate-buffered saline (PBS). (**e**) HaK survival after cooling and rewarming in PBS. (**f**) HEK293 survival after cooling and rewarming in PBS. All data presented as mean ± SD. * = *p* < 0.05, ** = *p* < 0.01; ANOVA post hoc Bonferroni. n.s. = non-significant.

**Figure 5 ijms-21-01864-f005:**
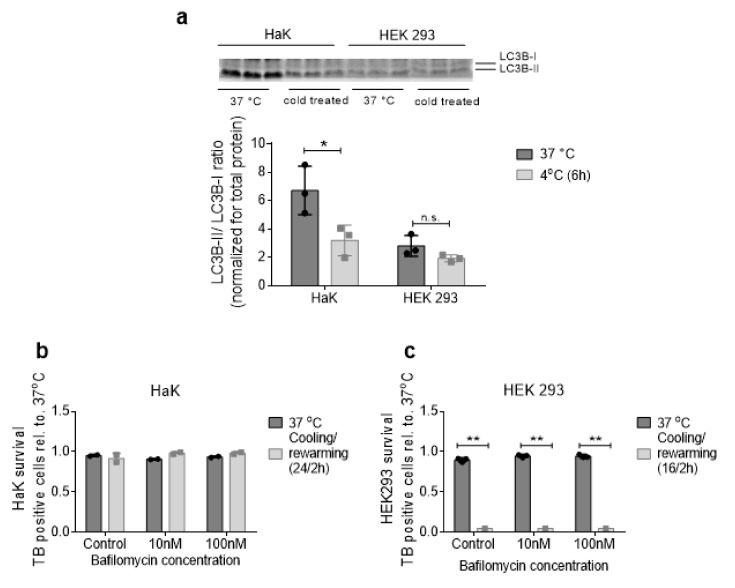
Autophagy activity in normothermic and hypothermic HaK and 293 cells. (**a**). Protein expression for LCB3-II and LCB3-I, expressed as LCB3-II/LC3B-I ratio. Normalized for total protein. Both in normothermic and hypothermic (4 °C, 6 h cooling) circumstances. (**b**) Survival after hypothermic in bafilomycin (autophagy inhibiter) treated HaK and (**c**) HEK 293 cells. All data presented as mean ± SD. * = *p* < 0.05, ** = *p* < 0.01; ANOVA post hoc Bonferroni. n.s. = non-significant.

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
