# Peer review of "Hibernator-Derived Cells Show Superior Protection and Survival in Hypothermia Compared to Non-Hibernator Cells"

_ijms, 2020, doi:10.3390/ijms21051864_

Round 1
Reviewer 1 Report
The article hibernator derived cells show superior protection and survival in hypothermia compared to non-hibernator cells is a well written article by Koen D.W. Hendricks et al. The authors here in the paper described elegantly the importance of mechanisms of hibernators used to sustain mitochondrial activity and inhibit the damage caused in hypothermic circumstances. Overall this is an interesting article and could be a good fit for the journal. The authors also show sufficient data to support their claims and hence I recommend for acceptance without any major review. I have a minor suggestions for authors below regarding Fig.5A.
The authors should show a house keeping gene like Beta actin or Vinculin in western blot as reference gene for equal protein normalisation.
Reviewer 2 Report
Panel D of Figure 1, I guess 37 should be 3 h. If true, please be careful your writing. Additionally, the panel label in the text should be similar to Figures such as A in the Figure but a in the text. Panel C & D of Figure 1, the HEK 293 cell viability at 48h was much lower than that of Hak cells, conflict with Figure1 in published paper. Please explain it. Line 104 to 106, “Consequently, hypothermia did not alter the baseline fold-increase in oxygen consumption of hibernator derived cells compared to non-105 hibernator cells.” Was not supported by Figure 2, please think it again. What degree to get Figure 2D? Why different panel was from different time point in Figure 2? Line 150 to 151, “H2O2 treatment did not affect MDA levels in HaK cells both in warm and cold conditions (Figure 3c).” needs more evidences because lots of published papers had shown that 500 uM of H2O2 treated HEK 295 cells resulted in much higher MDA level of HEK 293 cell if you search google using HEK293 cell and MDA level and H2O2 treatment. Results of 2.4: The results were got by culture cells in PBS only. It’s no effect on cell viability. I 100% doubted it. We usually use 1 x PBS to wash cells and it’s ok to keep cells for around 30 mins to 1 hour in PBS. But the common knowledge is that cultured cells diluted in PBS could be inadvertently lower the viability if automated measurement and can be consistently reproduced. If this study was confidence not effect of 500 uM H2O2 on cell viability, it’s big achievement and should deeply study it. Line 287: “HaK cells seems to recruit endogenous substrate(s) during cooling.” I can’t image that endogenous substrates were enough to support cell mitochondria generated ATP for 24 hours or more. Line 294 to 295: “possibly maintaining GSH levels, and prevention of excess ROS damage, thus precluding 294 ferroptosis”. There was no GSH data, please add this if you have to get the conclusion. For mitochondrial respiration assay, why this study didn’t give mitochondrial complex II substrate (Succinate) to determine oxygen consumption? Technically, every assay should have control to normalize. I don’t see what controls to be used for normalization for “Mitochondrial superoxides”, “mPTP opening”. If no control, the data were not believed.Author Response
Please see attachment

Round 2
Reviewer 2 Report
- I didn't see the response to my question 1. please reply for it.
- Response for my question 3, I am not happy to see the explanation because whatever you used unit; your tendency should be similar. Please think about and explain again.
- Response for my question 7, I disagreed with the response. 2-hour exposure to 500 uM H2O2 should be reflected by measurement of MDA level.
-
Response to my question 8, I disagree that “The interesting phenomenon is that HaK cells, specifically in hypothermic circumstances, survive in PBS”. How many percent of cells are survival? How to explain what nutrition support cell life?
